# RANKING COST: ONE-STAGE CIRCUIT ROUTING BY DIRECTLY OPTIMIZING GLOBAL OBJECTIVE FUNCTION

## ABSTRACT

Circuit routing has been a historically challenging problem in designing electronic systems such as very large-scale integration (VLSI) and printed circuit boards (PCBs). The main challenge is that connecting a large number of electronic components under specific design rules and constraints involves a very large search space, which is proved to be NP-complete. Early solutions are typically designed with hard-coded heuristics, which suffer from problems of non-optimal solutions and lack of flexibility for new design needs. Although a few learning-based methods have been proposed recently, their methods are cumbersome and hard to extend to large-scale applications. In this work, we propose a new algorithm for circuit routing, named as Ranking Cost (RC), which innovatively combines search-based methods (i.e., A* algorithm) and learning-based methods (i.e., Evolution Strategies) to form an efficient and trainable router under a proper parameterization. Different from two-stage routing methods ( i.e., first global routing and then detailed routing), our method involves an one-stage procedure that directly optimizes the global objective function, thus it can be easy to adapt to new routing rules and constraints. In our method, we introduce a new set of variables called cost maps, which can help the A* router to find out proper paths to achieve the global objective. We also train a ranking parameter, which can produce the ranking order and further improve the performance of our method. Our algorithm is trained in an end-to-end manner and does not use any artificial data or human demonstration. In the experiments, we compare with the sequential A* algorithm and a canonical reinforcement learning approach, and results show that our method outperforms these baselines with higher connectivity rates and better scalability. Our ablation study shows that our trained cost maps can capture the global information and guide the routing result to approach global optima.

## 1 INTRODUCTION

As described in Moore's Law (Schaller, 1997), the number of transistors in a dense integrated circuit (IC) increases exponentially over time and the complexity of chips and printed circuit boards (PCBs) becomes higher and higher. Such high complexity makes the IC design a time-consuming and error-prone work. Thus more capable automatic design systems, such as electronic design automation (EDA) tools, are needed to improve the performance. In the flow of IC designs, we need to find proper paths to place wires which connect electronic components on ICs, and these wires need to achieve expected connectivity under certain constraints. One of the most important constraints is that wires on the same layout should not intersect. In addition, to reduce the signal propagation delay, the wire-length should be minimized. This is a critical and challenging stage in the IC design flow (Hu & Sapatnekar, 2001), known as circuit routing, which has been studied by lots of researchers (Kramer, 1984; Zhang & Chu, 2012; He & Bao, 2020).

Circuit routing involves a large number of nets (a net is a set of vertices with the same electrical property) to be routed, which is computationally expensive and makes manual design extremely time-consuming (Kong et al., 2009; Coombs & Holden, 2001). Even under the simplest setting, where only two pairs of pins need to be routed, it is an NP-complete problem (Kramer, 1984). Although lots of circuit routing algorithms have been proposed (Zhang, 2016), there still remain

three major challenges: (1) Early solutions (Hu & Sapatnekar, 2001) are typically designed with hard-coded heuristics , which suffer from problems of non-optimal solutions (Zhang, 2016) and lack of flexibility over new design needs. Therefore, a more powerful routing method that does not depend on domain knowledge is highly desired. (2) To reduce the difficulty of complex routing problems, traditional routing algorithms often adopt a two-stage procedure — first global routing and then detailed routing (Chen & Chang, 2009; Kahng et al., 2011). The problem is that these two stages do not always coordinate well (Zhang & Chu, 2012; Shi & Davoodi, 2017). Sometimes a low-congested global routing result may lead to downstream detailed router un-routable. Hence, an end-to-end algorithm is preferred which can optimize the final global objective (e.g., the total wire-length) directly. (3) Although a few learning-based methods have been proposed (Liao et al., 2020; He & Bao, 2020) recently, their methods are hard to extend to large-scale applications. In real settings, there are lots of components and nets on a single chip, which shows greater demand for the scalability of routing algorithms.

To relieve the problems mentioned above, we propose a new algorithm, denoted as Ranking Cost (RC), for circuit routing. In this paper, we innovatively combine search-based methods (i.e., A* algorithm) and learning-based methods (i.e., Evolution Strategies (Salimans et al., 2017)) to form an trainable router with proper parametrization. Our method is flexible for integrating new constraints and rules so long as they can be merged into the global objective function. Moreover, our method is an one-stage algorithm, which optimizes the global objective function directly. In our method, we introduce a new set of variables called cost maps, which can help the A* routers to find out proper paths to achieve the global objective. We also train a ranking parameter, which can produce the ranking order and further improve the performance of our method. In the experiments, we compare our method with the commonly used A* method and a canonical reinforcement learning approach, and results show that our method outperforms these baselines with higher connectivity rates. Our ablation study also shows that trained cost maps can capture the global information and guide the routing solution to approach global optimal. Experiments also show that our method is scalable to larger applications.

## 2 RELATED WORK

In this section, we summarize the related work on circuit routing.

### 2.1 TWO-STAGE ROUTING ALGORITHMS

The routing problem can be heuristically separated into two stages, the first being the global routing step, followed by detailed routing. On the one hand, there are multiple heuristic-based approaches for global routing including regionwise routing (Hu & Sapatnekar, 2001), force-directed routing (Mo et al., 2001), and rip-up and reroute (Cho et al., 2007). On the other hand, the most commonly used detailed routing algorithms are channel routing and its variants (Ho et al., 1991; Mandal et al., 2020), which decompose the routing region into routing channels and generate wires in these channels (Chen & Chang, 2009). One main issue of the two-stage methods is that these two stages do not always coordinate well (Shi & Davoodi, 2017), which results in enormous difficulty in joint optimization. Instead, our method is an one-stage algorithm and new design constraints can be simply involved into the objective function without changing the algorithm itself.

### 2.2 SEQUENTIAL ROUTING ALGORITHMS

A more straightforward strategy for circuit routing is to select a specific order and then route nets sequentially, e.g., sequential A* algorithm and Lees algorithm (Huang et al., 2014; Malavasi & Sangiovanni-Vincentelli, 1993). The major advantage of this type of approaches is that the congestion information for previously routed nets can be taken into consideration while routing the current one. However, the drawback of these sequential approaches is that the quality of the solution is very sensitive to the orders (Zhang, 2016). Moreover, earlier routed paths only focus on finding their own best solutions and are impossible to take into account the situation of subsequent paths. Such greedy strategies may make a solvable circuit routing problem insolvable. Figure 1 shows an example that the sequential A* algorithm will fail to handle. In this example, there are two pairs of points to be routed, i.e., we should connect start vertices $S_i$ and end vertices $E_i, i \in \{1, 2\}$, respectively. If we

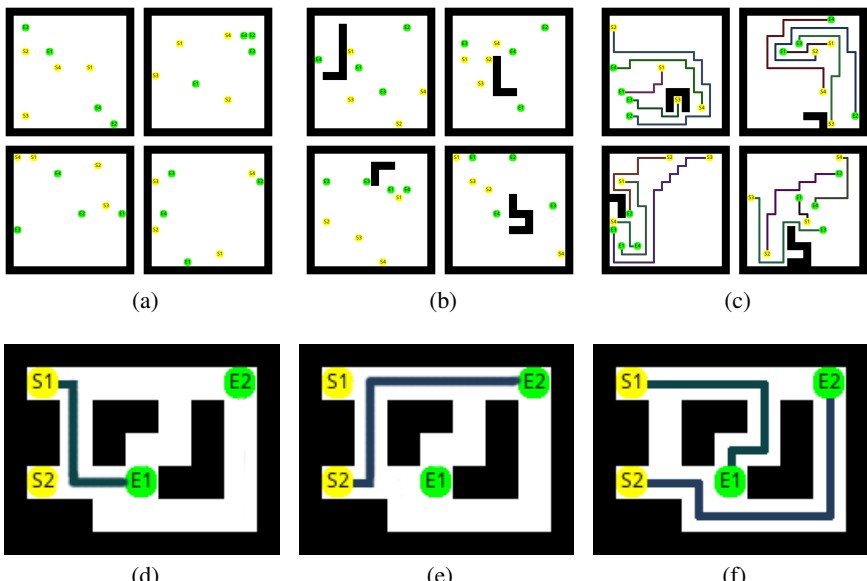

Figure 1: Yellow vertices are the start vertices, green vertices are the end vertices and black blocks are obstacles where paths can not pass through. (a) Examples of maps without obstacle.(b) Examples of maps with obstacles. (c) Circuit routing results derived by our algorithm. (d) and (e) show that sequential A* algorithm will greedily find a shortest path of one pair, and it fails to obtain a global solution. (f) shows the solution found by our algorithm, which is actually globally optimal.

connect the pair $(S_1, E_1)$ first in its shortest path, it will make $(S_2, E_2)$ disconnected as showed in Figure 1(d), and vice versa (Figure 1(e)). But this case can be solved easily by our algorithm as showed in Figure 1(f), which means that our algorithm can take into account the global information when generating each path.

## 2.3 LEARNING-BASED ALGORITHMS FOR CIRCUIT ROUTING

There are some learning-based algorithms for circuit routing (Qi et al., 2014; Liao et al., 2020). Early methods are more focusing on supervised learning, such as learning models to predict routing congestion (Qi et al., 2014) or predicting the routability (Zhou et al., 2015; Xie et al., 2018). But the performances of these supervised models are constrained by the limits of human knowledge. Recently, Liao et al. (2020) trained a deep Q-network (DQN) to solve the global routing problem and their method still suffers from the miscoupling problem of two-stage routing algorithms. In addition, He & Bao (2020) reform circuit routing to a tree search problem and utilize a deep neural network as the rollout policy. However, their rollout policy is trained in a supervised manner with artificial data. Instead, our method optimizes the global objective function directly and does not require any artificial data or human demonstration.

## 3 BACKGROUND AND NOTATIONS

In this section, we will formalize the circuit routing problem first and then give a brief introduction to the OpenAI-ES algorithm (Salimans et al., 2017).

## 3.1 CIRCUIT ROUTING

Circuit routing is a path search problem, where the goal is to find non-intersecting paths that connect an arbitrary number of pairs of vertices. It can be formalized as a grid graph $G = (V, E)$, where each vertex $v_i \in V$ represents an intersection on the grid, and each edge $e_{ij} \in E$ represents the path between $v_i$ and its 1-hop neighbors $v_j \in V$. And the non-routable obstructive vertices form a

obstacle set, denoted as $O \subset V$. In circuit routing, a net $N = \{v_{n1}, v_{n2}, ...\} \subset V$ is a set of vertices that need to be connected and a multi-vertex net can be decomposed into multiple two-vertex nets via a minimum spanning tree (MST) or a rectilinear Steiner tree (RST) (de Vincente et al., 1998; Hu & Sapatnekar, 2001). Following He & Bao (2020), we simplify this problem by letting each net only contain two vertices, i.e, a two-vertex net is defined as $N = \{v_s, v_e\}$ with a start vertex $v_s$ and an end vertex $v_e$. The net should be connected by a path $P = [v_1, v_2, ..., v_n]$ with $v_s = v_1$, $v_e = v_n$ and $P \cap O = \varnothing$. We use $|P|$ to represent the length of the path $P$. Given a set $\mathcal{N}$ of nets $\{N_i\}$, we need to find a set $\mathcal{P}$ of paths $\{P_i\}$ that connect these nets. A reasonable routing plan for a given set of nets is that all the paths do not share any vertex. In some cases, the routing problem do not have a solution because such a non-intersecting set $\mathcal{P}$ does not exist. In circuit routing, the routed paths should satisfy some constraints or achieve some goals. In this paper, we choose the most commonly used goal, i.e., minimize the total length of paths. Given a solution $\mathcal{P}$, the total length of paths is defined as $L = \sum_{P \in \mathcal{P}} |P|$.

## 3.2 EVOLUTION STRATEGIES

Evolution Strategies (ES) is a class of black box optimization methods (Rechenberg, 1994; Hansen & Ostermeier, 2001). Recently, an evolution strategy variant, referred as the OpenAI-ES (Salimans et al., 2017), has attracted attention because it could rival the performance of modern deep reinforcement learning methods on multiple control tasks. In the OpenAI-ES, there is a reward function, denoted as $F(\psi)$, where $\psi$ is a solution vector, which is sampled from a probability distribution function $\psi \sim p_\theta(\psi)$. The goal is to maximize the expected value $J(\theta)$, which is defined as:

$$\max_\theta J(\theta) = \max_\theta \mathbb{E}_{\psi \sim p_\theta}[F(\psi)] = \max_\theta \int F(\psi) p_\theta(\psi) d\psi. \tag{1}$$

A straightforward way to achieve this is to use the gradient ascent:

$$\theta \leftarrow \theta + \alpha \nabla_\theta J(\theta), \tag{2}$$

where $\alpha$ is the learning rate. In the OpenAI-ES, a score function estimator is used to calculate the gradient, which is similar to REINFORCE (Williams, 1992):

$$\nabla_\theta J(\theta) = \nabla_\theta \mathbb{E}_{\psi \sim p_\theta}[F(\psi)] = \mathbb{E}_{\psi \sim p_\theta}[F(\psi) \nabla_\theta \log p_\theta(\psi)]. \tag{3}$$

Usually, $p_\theta$ is an isotropic multivariate Gaussian with mean $\theta$ and fixed covariance $\sigma^2 I$. And the expected value can be rewritten as $\mathbb{E}_{\psi \sim p_\theta}[F(\psi)] = \mathbb{E}_{\epsilon \sim N(0,I)}[F(\theta + \sigma \epsilon)]$. Thus the gradient estimator changes to:

$$\nabla_\theta J(\theta) = \nabla_\theta \mathbb{E}_{\epsilon \sim N(0,I)}[F(\theta + \sigma \epsilon)] = \frac{1}{\sigma} \mathbb{E}_{\epsilon \sim N(0,I)}[F(\theta + \sigma \epsilon) \epsilon], \tag{4}$$

where $\epsilon$ is sampled from a standard normal distribution. Once we form an objective function $J(\theta)$, the gradient $\nabla_\theta J(\theta)$ can be approximated via Equation 4 and parameters $\theta$ can be updated via Equation 2.

## 4 METHODOLOGY

In this section, we will first introduce the Ranking Learning algorithm and then the Cost-Map Learning algorithm. At last, we give out our final algorithm (Ranking Cost) and its training strategy.

### 4.1 SEQUENTIAL A* WITH RANKING LEARNING

The A* algorithm can be used to find the shortest path in a two-vertex net $N = \{v_s, v_e\}$, which is considered more efficient than breadth-first search. A* finds a path between a start vertex $v_s$ and an end vertex $v_e$, with a priority function defined as:

$$s(v) = g(v) + h(v), \tag{5}$$

where $v$ is a vertex in the graph, $g(v)$ represents the total length from $v_s$ to the current vertex $v$, and $h(x)$ represents the future heuristic cost from $v$ to the end vertex $v_e$. The $h(x)$ is often set to

the Euclidean distance (in this work) or Manhattan distance. In a circuit routing problem, there are $k$ nets to be routed. Given a specific ranking order $R$, we can apply the A* algorithm to each net sequentially. For example, if there are 3 nets $\{N_1, N_2, N_3\}$ to be routed and a ranking order $R = (2, 1, 3)$ is given. We can apply A* to these 3 nets following the order $N_2 \rightarrow N_1 \rightarrow N_3$. After the routing procedure, we can get a total path length from this ranking order, denoted as $L(R)$. Different ranking orders will result in different total lengths. We can go through all the ranking orders and get their corresponding total lengths. The ranking order with minimum total length can be used as the final solution. However, this is unacceptable in time complexity because a routing problem with $k$ nets has $O(k!)$ different ranking orders.

To relieve the problem of combinatorial explosion of ranking orders, we propose a method to learn the ranking order. We define a $k$-dimension ranking parameter $\theta_r = \{\beta_1, \beta_2, ..., \beta_k\}$. And the ranking order is determined completely by the ranking parameter. More concretely, the order of values in $\theta_r$ is exactly the routing order, and the net with largest value will be routed first and the rest may be deduced by analogy. For example, given a ranking parameter $\theta_r = \{0.5, 0.2, 0.4\}$, we have $\beta_1 > \beta_3 > \beta_2$ and the corresponding ranking order will be $R_{\theta_r} = (1, 3, 2)$. And the ranking parameter will determine the final routing result and also the total path length. We define a reward function over the ranking parameter:

$$F(\theta_r) = -L(R_{\theta_r}), \tag{6}$$

where $L(R_{\theta_r})$ is the total length when taking the order $R_{\theta_r}$, and less length leads to larger reward. As described in Section 3.2, we define the expected value over $F(\cdot)$ as:

$$J(\theta_r) = \mathbb{E}_{\psi \sim p_{\theta_r}} F(\psi) = \mathbb{E}_{\epsilon \sim N(0,I)} F(\theta_r + \sigma\epsilon). \tag{7}$$

The gradient $\nabla_{\theta_r} J(\theta_r)$ can be estimated via Equation 4, and $\theta_r$ can be updated via Equation 2.

As mentioned in Section 2.2, a sequential routing algorithm may make a solvable circuit routing problem insolvable. And even a method with a learned ranking order will still suffer from this problem. In the following section, we will introduce how our method could overcome this problem by coupling the A* algorithm with cost maps.

## 4.2 CIRCUIT ROUTING WITH COST MAPS

As introduced in previous section, $g(v)$ in Equation 5 represents the total length from start vertex $v_s$ to current vertex $v$, thus the A* algorithm can only search for the shortest path of current net and it can not take into account the information of following paths. Our goal is to add global information to the A* routers so that these routers can cooperate together to achieve their common objective. First, for each net $N_i, i \in \{1, ..., k\}$, we define its cost map as $C_i = \{c_1, c_2, ..., c_m\} \in \mathbb{R}^m$, where $m = |V|$ is the number of vertices in graph $G$. Therefore, there are total $k$ cost maps, and all the cost maps can be learned with our algorithm. We apply a cost-map function $C(i, v_j) = c_j, c_j \in C_i$ to simplify the notation. In this way, we can reformulate the A* algorithm by adding the cost maps:

$$s(v) = g(v) + \sum_{i=1}^{k} C(i, v) + h(v), \tag{8}$$

where $g(v)$ and $h(v)$ are with the same definitions as in Equation 5. The cost-map enhanced A* algorithm reveals two key points: (1) When removing $g(v)$ and $h(v)$ from Equation 8, it will degenerate into a pure learning-based algorithm and the path will be determined completely by the cost function. (2) When removing the cost functions or set them to zero (i.e., $C(i, v) = 0$), it is a pure search-based algorithm. Hence, it is a method which combines the searched-based algorithm and the learning-based algorithm and could take advantage of both sides. On the one hand, the A* search makes our method easier to find a connected path in a complex environment, where learning-based methods are hard to find a connected path (Tamar et al., 2016). On the other hand, the global information can be merged into cost maps and the global objective can be approached by tuning the cost maps.

To learn the cost maps, we first define a cost-map parameter $\theta_c \in \mathbb{R}^{k \times m}$ and the value in cost maps are defined as $c_j = \max(0, \theta_c[i, j]), c_j \in C_i$. Given a ranking order $R$ and cost maps, we can route the nets sequentially using the cost-map enhanced A* and a total path length $L$ will be obtained from

the routing result. Similar to Equation 6, we define a reward function over the cost-map parameter and also its the expected value:

$$F(\theta_c) = -L(R, C_{\theta_c})$$
$$J(\theta_c) = \mathbb{E}_{\psi \sim p_{\theta_c}} F(\psi) = \mathbb{E}_{\epsilon \sim N(0,I)} F(\theta_c + \sigma\epsilon), \tag{9}$$

where $C_{\theta_c}$ presents the cost maps derived from $\theta_c$, and $\theta_c$ can be updated via Equation 2. By iteratively executing the A* search and cost-map learning step, our method could approach the optimal solution in the process of time and solve some problems that a sequential routing algorithm can not solve as showed in Figure 1. In next section, we will introduce how the cost-map learning can couple with ranking learning and how to train the algorithm efficiently in practice.

### 4.3 RANKING COST ALGORITHM

Our final algorithm, denoted as Ranking Cost (RC), can learn the ranking parameter $\theta_r$ and cost-map parameter $\theta_c$ jointly. In Equation 8, all the cost maps are used when calculating the priority function and the ranking order has no impact on the cost maps. To learn the cost-map with ranking order, we change the priority function to:

$$s_j(v) = g_j(v) + \sum_{i=j+1}^{k} C(i, v) + h_j(v), \tag{10}$$

where $s_j(v)$ is the priority function for the net whose ranking order is the $j$-th. This modification is reasonable, because the earlier routed nets will use more cost maps and observe more global information. For example, when routing the first net, its ranking index is $j = 1$ and it will use all the rest $k - 1$ cost maps. As a result, the first routed path focuses more on its impact on other unrouted nets. When routing the last net, its ranking index is $j = k$ and no cost map will be used. As a result, the last net is routed via a normal A* algorithm and the searched solution is actually the shortest path at current situation. As all the previous nets have been routed, the best way for routing the last net is just to find out the shortest path. Our RC algorithm achieves this naturally and makes it flexible for arbitrary ranking orders. When the ranking order changes, the order and the usage of cost maps change accordingly. In this way, the reward function can be defined as:

$$F(\theta_r, \theta_c) = -L(R_{\theta_r}, C_{\theta_c}), \tag{11}$$

and both $\theta_r$ and $\theta_c$ can be updated via the OpenAI-ES algorithm. Actually, the reward function $F$ can be integrated with arbitrary metrics, such as the signal latency. Thus the cost maps are free to adapt to new constraints and design rules.

In the OpenAI-ES algorithm, we will sample a fixed number of Gaussian noises and add them to original parameters to form new parameters. These new parameters are then fed into $n$ evaluators (or workers). The evaluator has an independent environment and executes the algorithm based on received parameters. Finally, each evaluator will return a scalar reward. To stabilize the training process, we normalize collected rewards $\{r_1, ..., r_n\}$ from all the evaluators:

$$\bar{r}_i = (r_i - r_{mean})/r_{std}, \tag{12}$$

where $r_{mean}$ and $r_{std}$ are the mean and standard deviation of all the rewards. Algorithm 1 in Appendix A shows the overall training procedure of our RC algorithm. In practice, all the evaluators can execute in parallel as proposed in Salimans et al. (2017). The source code of Ranking Cost can be found in the supplementary material.

## 5 EXPERIMENTS

In this section, we will compare our proposed method with some other baselines. We will also study how the ranking learning and cost maps work in our algorithm. Finally, we further show the scalability of our algorithm for larger applications.

### 5.1 METHODS EVALUATION

To evaluate our algorithm, we build a grid map simulator as the test environment. We construct 300 maps with three different sizes: $16 \times 16$, $32 \times 32$ and $64 \times 64$. These maps are split into two

| | $16 \times 16$ (4 pairs) | | $32 \times 32$ (6 pairs) | | $64 \times 64$ (10 pairs) | | time (s/map) |
|---|---|---|---|---|---|---|---|
| | no obstacle | with obstacles | no obstacle | with obstacles | no obstacle | with obstacles | |
| Seq A*$_{(5)}$ | 0.94(37.1) | 0.9(37.2) | 0.96(114.0) | 0.58(116.7) | 0.8(459.9) | 0.08(393.7) | 0.8 |
| Seq A*$_{(200)}$ | 0.96(36.7) | 0.96(**36.9**) | 1.0(**111.3**) | 0.78(**111.1**) | 0.84(457.2) | 0.24(383.0) | 41.5 |
| VINs | 0.84(37.1) | 0.86(37.2) | 0.54(129.4) | 0.18(122.2) | 0.0(-) | 0.0(-) | 1.4 |
| CL | 0.94(38.3) | 0.9(37.7) | 0.98(114.6) | 0.68(118.7) | **0.88**(485.4) | 0.16(408.3) | 37.5 |
| RC(ours) | **1.0**(36.7) | **0.98**(37.0) | **1.0**(113.4) | **0.82**(111.8) | 0.86(**431.3**) | **0.32(378.3)** | 38.3 |

Table 1: Evaluation results of different algorithms. Two values are reported. The first value is the success rate (the higher the better) and the second value in the bracket is the common average length (the lower the better). Ranking Cost achieves the best performance on all the maps. The comparison between CML and RC shows that the ranking parameter learning can improve the performance of our algorithm. We also report the wallclock time of different algorithms in the last column. The Ranking Cost algorithm takes more running time, but it is worthy to sacrifice more time to earn better solutions.

types, i.e., a simple one without obstacle and a more complex one with obstacles. Figure 1(a) and Figure 1(b) show some of the maps used in our experiments. Algorithms will be evaluated on these maps, and the success rates and the average of total lengths will be reported. More details about the maps can be found in Appendix B.

**Baselines**:
**Sequential A***: Sequential A* is a common used search-based algorithm in circuit routing. It routes nets with a specific ranking order and different orders will lead to different results. In the experiments, we randomly sample 5 different ranking orders and run the algorithm 5 times. The best score of 5 runs will be reported. We further randomly sample 200 different ranking orders and run the algorithm 200 times. The best score of 200 runs will be reported.
**Value Iteration Networks (VINs) (Tamar et al., 2016)**: The VIN is a learning-based approach for routing on grid maps. VINs use the value iteration from reinforcement learning and can achieve better performance compared with supervised models. We apply VINs to the circuit routing problem by sequentially executing it on each net. Similar to sequential A* algorithm, we randomly sample five different ranking orders and run the VIN five times. The best score of five runs will be reported.
**Cost-Map Learning (CML)**: We construct a Cost-Map Learning algorithm from the Ranking Cost. In the CML, the ranking parameter will not be trained (the ranking order is fixed) and only the cost-map will be trained.

The Ranking Cost (RC) as presented in Algorithm 1 is our final method for circuit routing problem and more details about its hyper-parameters can be found in Appendix C. Table 1 shows the evaluation results of different algorithms on all the maps. Two values are reported for each algorithm. The first value is the success rate (the higher the better) and the second value in the bracket is the common average length (the lower the better). Our final algorithm (Ranking Cost) achieves the best performance on all the maps. On the most complex maps (with size $64 \times 64$, obstacles and 10 pairs of vertices), Ranking Cost outperforms other baselines by a large margin, which implies that our method has greater advantage on handling more challenging tasks. The comparison between Cost-Map Learning and Ranking Cost shows that the ranking parameter learning can further improve the performance of our algorithm.

## 5.2 ABLATION STUDY

In this section, we study how each part of our algorithm works and the scalability of the RC.

### 5.2.1 IMPACT OF RANKING ORDERS

To show the impact of the ranking learning, we increase the sample number of orders for sequential A* and use the Ranking Learning algorithm described in Section 4.1 as comparison. Table 2 in Appendix D shows the changes of success rates with different order sample numbers and Figure 2(a) shows the curve of success rates and sample numbers. The result shows that we can improve the

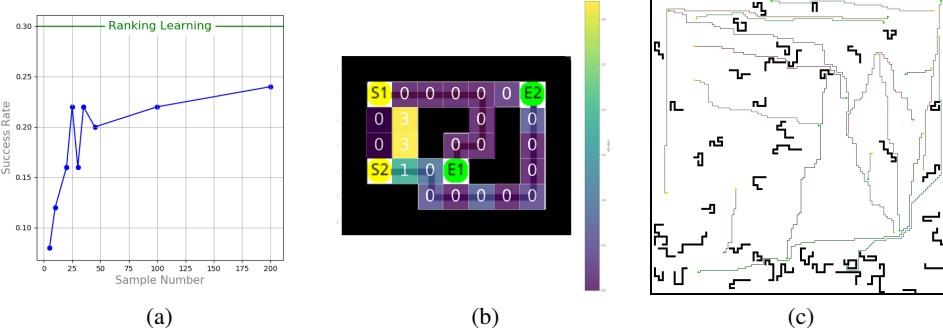

(a)  (b)  (c)

Figure 2: (a) The curve of success rates and sample numbers. Our learned ranking order achieves better performance than randomly sampled orders. (b) Trained cost map of the example from Figure 1(f). There are two larger cost values (the yellow grids) in the cost map, which will give out a large cost if the path chooses to go through here. As a result, the first pair $((S_1, E_1))$ will not take its shortest path which will block the second pair$((S_2, E_2))$. It shows that cost maps can capture the global information and guide routers to approach global optimal. (c) An example of routing results of our algorithm on maps with size of $150 \times 150$ and with 15 pairs, which implies that our method could be adapted to larger scales of applications.

success rate by increasing the order sample number, but it is hard to further improve the performance by linearly increasing the sample number as the order complexity is $O(k!)$, where $k$ is the number of pairs (or nets). However, our learned ranking order can achieve much better performance than randomly sampled orders.

### 5.2.2  IMPACT OF COST MAPS

To study the impact of cost maps, we fix the ranking order and only train the cost-map parameter, and we use the example map showed in Figure 1(f). We fixed the ranking order as $R = (1, 2)$ and train its cost maps. Because the ranking order is fixed and there are only two cost maps, only the second cost map will be used when routing the first pair (i.e., $(S_1, E_1)$). Finally, we visualize the trained cost maps as showed in Figure 2(b). There are two larger cost values (the yellow grids) in the cost map, which will give out a large cost if the path chooses to go through here. As a result, the first pair $((S_1, E_1))$ will take the path as showed in Figure 1(f) instead of taking its shortest path which will block the second pair$((S_2, E_2))$. It shows that our cost maps can capture the global information and guide routers to approach global optimal.

### 5.2.3  SCALABILITY OF RANKING COST ALGORITHM

In Liao et al. (2020), their algorithm is only evaluated on the maps with size of $8 \times 8$. In He & Bao (2020), their algorithm is evaluated on the maps with size of $30 \times 30$ and with only 5 pairs. The search space is huge in their methods, which prevent them from applying to larger maps. To test the scalability of our algorithm, we apply our algorithm on larger maps with size of $150 \times 150$ and with 15 pairs. Figure 2(c) shows the routing result of our algorithm, which implies that our method could be adapted to larger scales of applications.

## 6  DISCUSSION

In this paper, we propose a novel algorithm, called Ranking Cost, to solve the historically challenging circuit routing problem. In our method, we combine search-based algorithms (i.e., A*) and learning-based algorithms (i.e., evolution strategies) to form a new powerful algorithm which could take advantage of both sides. Our method is a one-stage circuit routing algorithm which can optimize the global objective function directly, and it is easy to implement and flexible for new design rules and constraints. Experimental results show that our method is powerful and scalable to more complex tasks. In the future, we will extend Ranking Cost to broader applications, such as pedestrian path prediction, multi-agent path finding and robot navigation.

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

## A    RANKING COST TRAINING

---
**Algorithm 1** Ranking Cost Training

---
**Initialize:** Graph $G = (V, E)$, vertex number $m = |V|$, nets $\{N_1, ..., N_k\}$, ranking parameter $\theta_r \in \mathbb{R}^k$, cost-map parameter $\theta_c \in \mathbb{R}^{k \times m}$, learning rate, $\alpha$, noise standard deviation $\sigma_r$ and $\sigma_c$, evaluator number $n$.
**for** $t = 1, 2, ...$ **do**
    **for** $i = 1, ..., n$ **do**
        Sample $\epsilon_i \sim N(0, I)$.
        Get sampled ranking parameter $\hat{\theta}_r$ and cost maps parameter $\hat{\theta}_c$ with:

$$\begin{aligned} \hat{\theta}_r &= \theta_r + \sigma_r \epsilon_i, \\ \hat{\theta}_c &= \theta_c + \sigma_c \epsilon_i \end{aligned} \tag{13}$$

        Construct ranking order $R$ and cost maps $C$ from $\hat{\theta}_r$ and $\hat{\theta}_c$.
        Route nets on graph $G$ with the ranking order and cost maps.
        Compute rewards $r_i = F(\hat{\theta}_r, \hat{\theta}_c)$ from routing results.
    **end for**
    Collect all scalar reward $\{r_1, ..., r_n\}$ from each evaluator.
    Compute normalized rewards $\{\bar{r}_1, ..., \bar{r}_n\}$ via Equation 12.
    Update ranking parameter $\theta_r$ and cost map parameter $\theta_c$ with:

$$\begin{aligned} \theta_r &\leftarrow \theta_r + \alpha \frac{1}{n\sigma_r} \sum_{j=1}^{n} \bar{r}_j \epsilon_j \\ \theta_c &\leftarrow \theta_c + \alpha \frac{1}{n\sigma_c} \sum_{j=1}^{n} \bar{r}_j \epsilon_j \end{aligned} \tag{14}$$

**end for**

---

## B    ENVIRONMENT SETUP

To evaluate our algorithm, we build a grid map simulator as the test environment. Algorithms are evaluated on randomly generated grid maps. In the experiment, we take use of three different maps sizes: $16 \times 16$, $32 \times 32$ and $64 \times 64$. We construct two types of maps, i.e., a simple one without obstacles and a more complex one with obstacles. All the start vertices and end vertices are also randomly generated in both simple maps and complex maps. Each map with size of $16 \times 16$ contain 4 start-end pairs, and 6 pairs for $32 \times 32$ and 10 pairs for $64 \times 64$. We randomly generate 50 maps for each size and each type and there are total 300 random maps. We report the success rate and the common average of total length of each algorithm. The success rate is the proportion of successfully connected maps to all random maps. Because different algorithm will have different connected maps, it is not fair to compare the average length over each algorithm's own successful maps. A algorithm with only one successful case may have lower average length than a algorithm with $100\%$ success rate. Thus we only report the average length over common successful maps of all algorithms. All the maps can be found in the supplementary material.

## C    DETAILS OF HYPER-PARAMETERS

For both Cost-Map Learning and Ranking Cost, the maximum training episode is $1000$, learning rate is $0.001$, the number of evaluators is $40$, the noise standard deviation $\sigma_r$ is $0.1$ and $\sigma_c$ is $0.1$. All the rewards returned from evaluators will be scaled to $[-1, 0]$. When there is not a successful connection, a reward of $-1$ will be given. Our code can be found in the supplementary material.

## D    RESULT OF RANKING ORDERS

| Sample Number | 5 | 10 | 15 | 20 | 25 | 30 | 35 | 40 | 45 | 50 | 100 | 200 | Ranking Learning |
|---|---|---|---|---|---|---|---|---|---|---|---|---|---|
| Success Rate | 0.08 | 0.12 | 0.16 | 0.16 | 0.22 | 0.16 | 0.22 | 0.20 | 0.20 | 0.22 | 0.22 | 0.24 | 0.30 |

Table 2: The changes of success rates with different order sample numbers. The result shows that it is hard to improve the performance by linearly increasing the sample number. The learned ranking order can achieve much better performance than randomly sampled orders.

## E    OTHER ROUTING TASKS

There are also some other routing problems, such as routing of hydraulic systems (Chambon & Tollenaere, 1991), routing of ship pipes (Kang et al., 1999), routing of urban water systems (Christodoulou & Ellinas, 2010; Grayman et al., 1988), and routing of city logistics (Barceló et al., 2007; Ehmke et al., 2012). Multi-agent path finding, a task similar to circuit routing, has been exhaustively studied in robotics and video games (Silver, 2005; Sturtevant, 2014; Babayan et al., 2018; Atzmon et al., 2020). But in a multi-agent path finding task, paths are allowed to intersect as long as the agents do not appear in the same place at the same time. To a certain degree, circuit routing is more difficult than multi-agent path finding tasks.

