# OpenReview forum: "Ranking Cost: One-Stage Circuit Routing by Directly Optimizing Global Objective Function"
_ICLR.cc/2021/Conference — Reject_

### Official Review · AnonReviewer2 · 2020-10-18
**An interesting application of evolution strategies on a combinatorial problem, but the evaluation is limited.**

**Rating:** 5
**Confidence:** 3

**Review:**

### Summary

Author considers the circuit routing problem, where each net contains two nodes. The exact circuit routing problem is NP-hard in general, and author proposed to an approximate method. It uses several parameters to control the solution returned by a greedy search, i.e. the order of the nets to be routed and the greedy route selected by each net.  At last, evolution strategy is deployed to find the best parameter that results in the most optimal greedy solution. Author evaluated the method on multiple grid environments, and shows that the method performs better in terms of both accuracy and scalability compared to other similar methods that uses learning to find a good solution.

### Strength:
The paper is organized very clearly, and all the ideas are demonstrated with good explanations and examples.
I found the idea to be very simple and interesting. The algorithm seems to be quite easy to be implemented.

### Weakness:
I am concerning on the novelty of the paper. The paper uses parameter to control the execution of a sequential A* algorithm, and the parameters are updated using neural strategies. It is not clear why a sequential A* algorithm is selected here. For example, under what conditions of the routing problem, the coupling of evaluation strategy and the execution of a sequential A* algorithm can return an optimal solution.

On the significance part, I am not satisfied by experiments that author designs.
* On the baselines: Currently author only selects baselines that utilizes learning. As the circuit routing is a well known problem, author should include some baselines using traditional methods to demonstrate the significance of deploying RC on the routing problems.
* On the problem instances: Currently, author only considers the circuit routing in a grids with non-overlapping constraint. These problems can also be solved using traditional methods. Author can consider to evaluate RC on problems whose design constraint cannot be handled using traditional methods, as promised by the introduction. This can tighten the argument and reasonably narrow audience's focus on only the learning based algorithm.

### Comments:
* Have author tried to apply the idea of neural evolution on general combinatorial optimization problem,  e.g. approximate a Max-SAT problems? In particular, you can also learns a variable order and value order using the same formulation in RC. As many combinatorial problems can be reduced to Max-SAT, it potentially can increase the significance of the work.

### Questions:
* I found the problem is very similar to the disjoint vertex path problems. Is there any connection between the two problems?
*During training, whether the learned ranking can be decoupled with the learned cost map? For example, can you still obtain a good routing plane using the same cost map while changing to a different order that is suggested by algorithm? What is the cost map for the first net in Figure 1f)? Does it shows any interesting structure so that you can still obtain a good routing plan if we try to route the second net before the first one?

---

> ### Author Response · Authors · 2020-11-18
> **Thanks for your review**
>
> We sincerely thank you for your comprehensive comments on our paper and we carefully answer each of your questions as below.
>
> [related to vertex-disjoint path problems]
>
> Thank you for pointing out the related work. The circuit routing task is indeed a vertex-disjoint path problem. In the research community of the vertex-disjoint path problem, researchers have proved that it is an NP-complete problem, and they also add some constraints on the graph to reduce the difficulty when designing heuristic algorithms. However, our method is a learning-based method, and it can be used on general graphs.
>
> [order of the cost maps]
> 1. On the one hand, the order parameter and the cost map parameter are trained jointly, so the order and cost map are conditioned on each other, which means they are not statistically independent. A cost map with the wrong order could impact the final performance. For example, if we have two pairs to be routed and the order is fixed as (1,2), the first cost map will completely not be trained and only the second cost map will be trained according to Equation (10). If the order is changed to (2,1) after training, then the trained cost map (the second one) will not be used, which may lead to a bad result. When the trained order and cost maps are coupled, you shall not change the order if you want to get the best result.
> 2. On the other hand, the cost maps also have the ability to generalize to different orders. For example, if we have two pairs to be routed and the orders are not fixed, then both of  the orders,  (1,2) and (2,1) will occur in the learning procedure. Thus, both the first and the second cost map will be well trained. When testing with the order (1,2), only the second cost map will be used and vice versa. The generalization ability of cost maps over different orders makes the training of the order parameter possible, where the order will be ceaselessly changed during the learning procedure.
>
> [extend to other combinatorial optimization problems]
>
> In this work, we mainly focus on designing algorithms to solve the circuit routing problem. Indeed, our method has shown some insights for solving more general combinatorial optimization problems (e.g., we can also learn a variable order and values using the same formulation in RC for the Max-SAT problem), and we think it is worthy to be explored in the future work.
>
> [the reviewer states this problem can be solved by traditional algorithms]
>
> The circuit routing problem is an NP-complete problem, we don't find there is a universal traditional algorithm that can solve this problem in acceptable time complexity.
>
> [why a sequential A* algorithm is selected here]
>
> The A* search algorithm is one of the most efficient shortest path algorithms. And the sequential A* is the most commonly used algorithm in circuit routing tasks. That is why we choose the A* algorithm as our low-level planner.

---

### Official Review · AnonReviewer4 · 2020-10-25
**Review for Submission 45**

**Rating:** 6
**Confidence:** 3

**Review:**

##########################################################################
Summary:

In this paper the authors attempt to solve the circuit routing problem through a novel approach of combining search-based routing techniques (e.g. A* search) and evolutionary strategies (e.g. OpenAI-ES). The authors define new parameters e.g. cost maps and a ranking parameter which improve the efficiency and effectiveness of their circuit routing solution. As opposed to heuristics based (2-step) routing solutions, the authors develop a solution which optimizes a single global cost function. This makes the solution easily scalable to new constraints and designs while avoiding sub-optimal solutions (common problem for 2-stage routing). The authors show that their approach doesn’t require human demonstrations or training data of any kind thereby giving their solution an edge over learning-based techniques requiring training data. Lastly, through experiments the authors establish that their approach is scalable to larger applications.


##########################################################################
Reasons for score:

The authors present an all-round discussion for the different approaches to solve the circuit routing problem with their limitations. They also establish how the proposed approach excels where the previous approaches get stuck or give sub-optimal solutions. However, the authors do not present any information regarding the time complexity of their proposed approach. Along with that, the paper is also missing comparisons with the exact solutions proposed in past papers targeting this problem. Supplementary data and experiments can help alleviate these concerns.

Updated: The authors added the requested timing complexity data and additional experiments with better baselines to compare the proposed RC algorithm against. They present valid issues with reproducing some of the previous work. While a theoretical proof establishing the worst-case time complexity of RC to be better than random sampling plus Seq A* would be ideal. The empirical data presented does support the claim that RC algorithm is useful for finding more optimal solutions for large maps faster than Seq A* plus random sampling.

##########################################################################
Pros:

1.	The authors present a detailed description of existing approaches to solve the circuit routing problem along with the key limitations of the approaches discussed.
2.	Specifically, since the proposed approach does not require training data or human demonstration, it has a big advantage on other learning-based methods. Since, training data for routing problems is hard to generate (time-consuming) or find easily (lack of open source benchmarks/data).
3.	 Section 4 presents a well written incremental description of the proposed routing approach starting from ranking learning to cost map learning to ranking cost.
4.	The experiments presented in Section 5, clearly demonstrate the superiority of the proposed approach over the baseline approaches considered in the paper.
5.	Lastly, the ablation studies clarify the limitations of scaling the random sample bucket for rank learning. Figure 2 b) does a particularly good job of illustrating the usage of Cost maps. Further, the authors demonstrate the scalability of their approach to larger applications. Which are missing in the cited learning-based previous approaches.

##########################################################################
Cons:

1.	The presented approaches rely heavily on the ability of OpenAI-ES algorithm to learn the cost maps and ranking order that will help achieve optimal routing. It would be useful if the authors could present an analysis of the limitations of this algorithm. Are there any cases when it fails to converge to the optimal solution? Are there other alternative solvers that could replace this evolutionary strategy in such scenarios?
2.	Section 5, the choice of baseline approaches while covering the fundamental types of routing approaches (sequential, learning (RL) based, cost-map learning) do not show performance compared to the related works cited in the paper. For instance, Liao et al. (2020), He and Bao (2020), or 2-stage routing algorithms.
3.	While the authors correctly list the limitations of heuristic based (2-stage) routing algorithms, they fail to credit their simplicity which can be critical when routing very large-scale designs like VLSI chips. Such a tradeoff of simplicity vs optimality of solution would require a comparison of the time complexity of the different approaches analyzed in this paper and how they scale with the complexity of the problem.
4.	Albeit the heuristic based approaches might converge on sub-optimal solutions. However, if they are much faster than ranking cost, combining multiple trials of heuristics-based approaches with human expert interventions could give ranking cost a competition in finding the optimal routing faster.
5.	Above suggested analysis of time complexity will help understand up to what number of random samples of ranking orders can be tried, while matching the time taken by ranking cost, for Sequential A* and VIN baselines. Thereby allowing for a fairer comparison in the results presented in Table 1.

##########################################################################
Questions during rebuttal period:

1.	Kindly address the concerns regarding time complexity of the proposed and past routing approaches as described in the Cons section.

#########################################################################
Some typos:

(1)	Section 2.3, “Recently, Liao et al. (2020) trained* a deep Q-network”.
(2)	Section 6, “Our method is a* one-stage”.

---

> ### Author Response · Authors · 2020-11-18
> **Thanks for your review**
>
> We sincerely thank you for your comprehensive comments on our paper and we carefully answer each of your questions as below.
>
> [are there other alternative solvers that could replace this evolutionary strategy?]
>
> We think other black-box optimization methods, such as the REINFORCE algorithm, could also be used in our framework. But in the OpenAI-ES paper, authors reported that OpenAI-ES rivalled the performance of modern deep reinforcement learning methods on multiple control tasks. Thus, we choose OpenAI-ES as our optimization method.
>
> [time complexity of our algorithm]
>
> For learning-based algorithms, a common way to compare the time complexity is reporting the learning steps or wallclock time. We report the wallclock time of different algorithms in our new version.  It is obvious that the Ranking Cost algorithm takes more running time, since it involves in a learning procedure when solving each task. But we argue that the time consumption is acceptable, since our algorithm only takes less than one minute to get the solution. It is worthy to sacrifice a little more time to earn better solutions.
>
> [no comparison to Liao et al. 2020, He & Bao, 2020]
>
> It is impossible for us to implement the methods of Liao et al. 2020 and He & Bao, 2020, because their papers lack detailed method descriptions and fail to release related materials (code and datasets).  For example, He & Bao, 2020 labelled a private dataset and trained a supervised model, but they did not public their dataset and failed to describe the network structure. These papers lack key information and related materials, which prevents other researchers from re-implementing and following their work. Instead, we will public our circuit simulator, detailed source code and the full dataset, if accepted.
>
> [some typos]
>
> Thank you for helping us improve the paper. We have corrected these typos in the new version.

---

> > ### Comment · AnonReviewer4 · 2020-11-22
> > **Useful Updates, More Data could help further**
> >
> > I appreciate the authors’ detailed responses to some of the questions posed in my first review. Particularly, appreciate the authors finding out REINFORCE, an alternative solver that could substitute OpenAI-ES thereby making Ranking Cost approach agnostic of the evolutionary solver being used.
> >
> > The addition of wallclock times comparing the different routing algorithms is very useful. However, it also highlights that the time consumed by Ranking Cost for routing the same problem is much higher than the baselines considered. Such a skew makes the comparison unfair, with the possibility of higher wallclock time (no. of random samples) enabling the baselines to perform similar to or better than Ranking Cost. For e.g., Seq A* algo with 5 random trials spends 0.6 s/map compared to 38.3 s/map by Ranking Cost. Therefore, Seq A* algorithm with 300 (~ 5 x (38.3/6)) samples would actually be a fairer baseline. Which might bring down their common average length and increase success rate. Keeping this in mind, the ablation study presented in Section 5.2.1 seems incomplete too, as it only samples a max of 30 random ranking orders. Whereas, if an equal wallclock time permits higher number of samples (~100s), we might get comparable/better success rate than Ranking Cost. The authors should consider modifying their random sample numbers for their baselines to better match the proposed algorithm, without which it’s hard to assess the efficacy of Ranking Cost over baselines.
> >
> > Conceptually, Ranking Cost algorithm appears to converge to more optimal routing paths. However, the concern is whether its timing complexity is too high for the improvement it shows in success rate and average path length.
> >
> > Lastly, the authors present compelling arguments as to why they cannot reproduce past work from He and Bao (2020), however they do not present any data for heuristic based 2-stage techniques which are supposed to be worst performing algorithm. It would be useful if the authors could share data comparing Ranking Cost to such a baseline.
> >
> > Small Corrections:
> >
> > 1.	In the Abstract and Related Work sections, the authors claim that since the proposed solution is optimizing a one-stage global function it is more flexible to new constraints than 2-stage heuristic-based methods. However, the authors do not elaborate any constraints that the 2-stage heuristic-based methods cannot accommodate. Kindly provide at least one example of the same.
> > 2.	Section 4.1, “Different ranking orders will result in different paths*.” OR “Different ranking orders will result in different total lengths*”
> > 3.	Section 5.1, “Similar* to sequential A* algorithm, we randomly sample …”
> > 4.	Section 5.2.2, “We take* the example map …” OR “We use* the example map …”

---

> > > ### Author Response · Authors · 2020-11-23
> > > **Thanks for your new comments**
> > >
> > > We sincerely thank you for your new comments on our paper and we carefully answer each of your questions as below.
> > >
> > > [more runs on the sequential A* algorithm]
> > >
> > > We conduct 200 runs for the sequential A* algorithm in our new version. Results show that our method still outperforms these baselines.
> > >
> > > [ablation study in Section 5.2.1 needs more runs]
> > >
> > > We add more runs for the sequential A* algorithm in Section 5.2.1. Results show that our Ranking Learning algorithm still has better performance.
> > >
> > > [no comparison to two-stage methods]
> > >
> > > We tried but were not able to implement a two-stage baseline in the rebuttal period, but will gladly include it by the camera-ready. Even given that, we point out: the two-stage method is just a routing framework, which divides a complex routing task into two simpler routing stages. Any one-stage routing method, such as the sequential A* or our Ranking Cost algorithm, can be merged into a two-stage routing framework. Moreover, the sequential A* algorithm is the most commonly used algorithm in current EDA tools for circuit routing. Therefore, we ( and also recent learning-based papers) mainly compare algorithms with the sequential A* algorithm. We think the sequential A* algorithm can serve as a fair baseline in the circuit routing task.
> > >
> > > [new typos]
> > >
> > > Thank you for helping us improve the paper. We have corrected these typos in the new version.

---

> > > > ### Comment · AnonReviewer4 · 2020-11-23
> > > > **New results strengthen the benefit of RC algorithm**
> > > >
> > > > The new data added by the authors with higher number of random samples for the Seq A* baseline is very useful. As seen in Table 1, it corroborates the possibility that Seq A* approach provided with enough number of samples for a given circuit routing problem could match/outperform RC algorithm (e.g., 32x32, 6 pairs, no obstacle). However, it also highlights the superiority of RC routing for larger maps (e.g., 64x64, 10 pairs). The similar runtimes for Seq A* (200), justify the comparison with the baseline better than past results. They also help resolve concerns regarding trading off too much time complexity for optimal results.
> > > >
> > > > While it’s unclear that the authors completely address the concerns raised by R3 about the 40 parallel works used for RC algorithm, a key advantage of RC emerges clearly. RC enables a systematic survey of the routing design space (which grows with combinatorial complexity) to find optimal routing solutions more efficiently than random sampling. As the map size and number of nets to be mapped continues to grow larger, the number of random samples required for Seq A* to match RC would keep rising. Additionally, starting the random sampling with a different seed could vary the time to find optimal routing solution significantly.
> > > >
> > > > Building on the above observation, if the authors could attempt to derive some worst-case time-complexity estimates for RC vs Seq A* w/ random samples for a N x N map with M samples, then that would further lend credibility to my previous assertion. I understand this might not be trivial and might be pursued as future work.
> > > >
> > > > The authors also updated the ablation study conducted in Section 5.2.1 to include 200 samples for Ranking Learning algorithm and demonstrate the superiority of RC. It would be great if the authors added details like map size, number of pairs and obstacle/no-obstacle constraints for this analysis.

---

### Official Review · AnonReviewer1 · 2020-10-27
**Do the learned policies generalize to unseen board configurations?**

**Rating:** 5
**Confidence:** 4

**Review:**

The paper considers the problem of determining efficient routes for connecting pairs of source, destination points over a circuit board. Due to physical constraints no two routes may intersect, and the routes may not pass through obstacles if there are any. For efficient performance it is desirable that the paths are as short as possible; as such the authors consider minimizing the sum length of all the paths as the objective. The paper proposes a ranking cost algorithm that combines A* search with an evolution optimization technique for learning efficient routes. Experimental evaluations show the proposed approach outperforming an RL based approach.

Routing in circuit boards is an important problem for the vlsi community. Moreover, the problem has a combinatorial optimization flavor to it, which have been considered and is of broad interest to the iclr community as well. The main issue I have with the paper is the lack of sufficient motivation given for the proposed solution. E.g., why A* search, why OpenAI-ES? To me the proposed approach feels a bit “non-standard”, which leaves me wondering why the authors chose to go via this route. Could you have solved this via an RL formulation in one stage for example?

The other issue I have is regarding generalization. Can the parameters learned by the proposed algorithm be used for finding routes in unseen circuit board (or nets) configurations? As far as I understand, both training and evaluation is done for a fixed board and nets configuration. If this is indeed the case, then how is the learning approach justified over well-known black-box optimization (e.g., simulated annealing, genetic algorithms etc.) or fast approximation algorithms (e.g., integer programming with time cut-off etc.)?

Minor: is the variance parameter $\sigma$ in AI-ES learned as well, or is only the mean learned?

---

> ### Author Response · Authors · 2020-11-18
> **Thanks for your review**
>
> We sincerely thank you for your comprehensive comments on our paper and we carefully answer each of your questions as below.
>
> [why A* search, why OpenAI-ES]
> 1. The A* search algorithm is one of the efficient shortest path algorithms. And the sequential A* is the most commonly used algorithm in circuit routing tasks. That is why we choose the A* algorithm as our low-level planner.
> 2. To train the cost map parameters, we need a powerful black-box optimization method. Actually, OpenAI-ES is a black-box optimization method and rivals the performance of modern deep reinforcement learning methods on multiple control tasks (for evaluative results, please check the OpenAI-ES paper). Thus, we choose OpenAI-ES as our optimization method.
>
> [is the parameter $\sigma$ learned?]
>
> In the OpenAI-ES algorithm, this parameter is fixed and will not be learned. So we also treat it as a hyper-parameter in our work.
>
> [compared with other black-box optimization methods]
>
> Yes, we agree that it is ok to use different black-box optimization methods to optimize the parameters. And the OpenAI-ES has shown excellent performance on complex tasks and has better scalability with computing resources. Thus OpenAI-ES is more suitable for computation-dense tasks and this is the reason why we utilize the OpenAI-ES as the underlying black-box optimization method.

---

### Official Review · AnonReviewer3 · 2020-10-27
**Interesting and well written paper but experiments can be improved**

**Rating:** 5
**Confidence:** 3

**Review:**


	1.  Summary & contributions
This paper develops an approach for solving a (simplified) circuit routing problem, which is to find non-intersecting paths between pairs of points (so called nets) in a grid. The paper uses a learning based approach based on OpenAI Evolution Strategies (OpenAI-ES) which learns 1) the order in which to route the different nets and 2) so called cost-maps for each net which provide additional guidance to the A* algorithm to account for later nets while routing earlier nets (i.e. to 'avoid the greedy pitfall').

	2. Strengths & weaknesses
The paper is well written and generally easy to follow and uses a principled and elegant approach to solving an interesting problem. I especially like the 'natural' way in which a learned policy integrates with A* search, and the resulting desirable properties. This makes the method a very principled approach.

The major limitation of this work is that the proposed method is only compared in terms of solution quality, and not by computational cost. From the specified parameters (up to 1000 training iterations with 40 parallel workers), the necessary compute seems orders of magnitude higher than the baselines, for example the 'random approach' which takes the best of 5 tries. To substantiate the claim that the method "outperforms baselines with higher connectivity rates and better scalability", I think at least baselines should be evaluated with the same computational budget, and ideally the trade-off between solution quality and computational cost should be explored.

Also, I wonder, why is the method not compared to Liao et al. 2020, He & Bao, 2020?

Another aspect which is relevant is that, if I understood correctly, the 'policy' consist only of the ranking parameters and the cost maps, and is thus tied to a specific instance of the circuit routing problem which cannot generalize to new instances, such that the training procedure should be run again for each new instance, adding to the high cost.

	3. Recommendation
My current assessment is that the paper is marginally below the acceptance threshold.

	4. Arguments for recommendation
The paper is of high quality, well written and well motivated, but the experiments lack a comparison of the algorithm and baselines with respect to computational cost, which is relevant to support the claim that this method outperforms baselines and has better scalability.

	5. Questions to authors
Could you add experimental results with respect to running time or number of iterations, for example extend table D to a number of samples (40K ?) such that the running time is comparable with the proposed method?

	6. Additional feedback
Minor comments/suggestions/compliments/questions:
- The term 'score function J(theta)' can be confusing since typically the score function refers to grad log p_theta. Maybe 'objective function' could be an alternative?
- Why does A* use Euclidean distance as heuristic and not Manhattan distance which seems more sensible in a grid?
- I like the qualitative comparison which only compares length for problems solved by all solvers
- What is the key that makes this approach more scalable compared to  Liao et al. 2020, He & Bao, 2020?

---

> ### Author Response · Authors · 2020-11-18
> **Thanks for your review**
>
> We sincerely thank you for your comprehensive comments on our paper and we carefully answer each of your questions as below.
>
> [add time comparation]
>
> We added the wallclock time of different algorithms in the experiment in the new version. It is obvious that the Ranking Cost algorithm takes more running time, since it involves a learning procedure when solving each task. But we argue that the time consumption is acceptable, since our algorithm only takes less than one minute to get the solution.  It is worthy to sacrifice a little more time to earn better solutions.
>
> [about the "score function" $J(\theta)$]
>
> We have taken the reviewer's advice, and changed it to the "objective function" in the new version.
>
> [why Euclidean distance not Manhattan distance]
>
> It is optional to use Euclidean distance or Manhattan distance in our algorithm. Since they are just heuristic distance functions and neither of them can return the true distance, both methods can be chosen for the A* algorithm and they will not change the final result.
>
> [What is the key that makes this approach more scalable compared to Liao et al. 2020, He & Bao, 2020]
> 1. We think the key reason is that our method takes advantage of an efficient search-based method (the A* algorithm) and combines it with a learning-based method. Our method is built on top of the A* algorithm, and our learning part focuses on optimizing the parameter used by the A* algorithm.
> 2. However, Liao et al. 2020 and He & Bao, 2020 try to design algorithms to produce actions at each time step. Formally, they treat this task as a sequential decision-making problem, thus the search space grows exponentially with the running length (or the map size), which prevents their method from applying to larger maps.
>
> [no comparison to Liao et al. 2020, He & Bao, 2020]
>
> It is impossible for us to implement the methods of Liao et al. 2020 and He & Bao, 2020, because their papers lack detailed method descriptions and fail to release related materials (code and datasets).  For example, He & Bao, 2020 labelled a private dataset and trained a supervised model, but they did not public their dataset and failed to describe the network structure. These papers lack key information and related materials, which prevents other researchers from re-implementing and following their work. Instead, we will public our circuit simulator, detailed source code and the full dataset, if accepted.
>
> [can't generalize to unseen tasks]
>
> Our algorithm should be trained on each task independently and can't generalize to unseen tasks. We also point out:
> 1. Each circuit routing task is difficult enough, i.e., it is an NP-complete problem, so it will be challenging to design a single model to solve all the tasks.
> 2. In reality, there are various sizes and shapes of circuit boards. It will be challenging to design a single model to take different kinds of boards as inputs.
>
> Designing a model with generalization ability will be explored in our future work.

---

> > ### Comment · AnonReviewer3 · 2020-11-21
> > **Appreciate added timings but not enough to support claims of 'outperforming baselines'**
> >
> > Let me start by saying that I sympathize with the situation of lack of details and code from previous work which makes comparison difficult and appreciate that you intend to publish code.
> >
> > Thanks for adding the run times to the paper. This is a great first step but I do not think it is not sufficient to support the claim that the ranking cost method 'outperforms baselines and has better scalability' as stated in the abstract. To elaborate:
> >
> > From the result, it appears sequential A* is faster but Ranking Cost gives better results. Based on this, it is not possible to claim superiority of one method over the other. For a fair comparison, at least both algorithms should be compared given equal runtime, but better would be to compare the full runtime vs quality curve if both algorithms allow to make this trade-off, e.g. by varying the number of sampled ranking orders / trained episodes.
> >
> > The seq. A* baseline uses 5 samples and has a runtime of 0.6 seconds, so I would assume it can do 300 samples in 36 seconds and this would be a much more fair comparison to the Ranking Cost Algorithm. Additionally, if I understand correctly the Ranking Cost algorithm uses 40 parallel workers and the seq. A* baseline does not, so this should also be accounted for in a fair comparison of the timing. I wonder what the results of seq. A* would be with 40*300 = 12,000 samples.
> >
> > As I initially stated, I do like the method as well as the paper in general, but I do not see the major claim of 'outperforming the baselines' supported, which makes that I cannot recommend to accept this paper.

---

> > > ### Author Response · Authors · 2020-11-23
> > > **Thanks for your new comments**
> > >
> > > We sincerely thank you for your new comments on our paper and we carefully answer each of your questions as below.
> > >
> > > [more runs on the sequential A* algorithm]
> > >
> > > We conduct 200 runs for the sequential A* algorithm in our new version. Results show that our method still outperforms these baselines.

---

> > > > ### Comment · AnonReviewer3 · 2020-11-23
> > > > **What about 40 workers?**
> > > >
> > > > I appreciate the extra experiments which makes a much stronger baseline. While this is already more fair, if I understand correctly, it does not make use of 40 parallel workers. I think taking 40 x 200 samples in parallel would make a truly fair comparison. Could the authors comment on this?
> > > >
> > > > As a small comment, why does 200 samples take longer than 40x 5 samples (40 * 0.8s = 32s), and has the runtime for 5 samples increased from 0.6 to 0.8 s?

---

> > > > > ### Author Response · Authors · 2020-11-23
> > > > > **about 40 workers and running time**
> > > > >
> > > > > [about 40 workers]
> > > > >
> > > > > In each step of sequential A*, the sampled order is totally different from each other. However, in the RC algorithm, orders can be the same for all the workers. So, using 40 workers for RC does not mean that it has the 40 times number of samples to the A* algorithm.
> > > > >
> > > > > [why 0.06 to 0.08]
> > > > >
> > > > > Because we re-run the A* algorithm and update the running time. We don't have a clean server (many people can use the server at the same time), so the wallclock time could be affected by lots of factors.

---

### Author Response · Authors · 2020-11-18
**Overall Response**

We thank the reviewers. Though final ratings are negative, the reviews themselves sound positive.

Our paper "**is of high quality, well written and well motivated**" (R3). Our work "uses a principled and elegant approach to solving an interesting problem" (R3), and our idea is also "very simple and interesting" (R2).  Our paper "is organized very clearly" (R2) and has "a well-written incremental description of the proposed routing approach" (R4).

The primary concern is that our work is "weak" in the experiment, which we humbly disagree with (more below).
1. Recent papers (Liao et al. 2020, He & Bao, 2020), only compare their method with sequential A*. Instead, our method not only compared with sequential A*, but also compared with a deep reinforcement learning approach (VINs).
2. We don't take Liao et al. 2020 and He & Bao, 2020 as baselines, because we are unable to implement their methods.  The main reason is that their papers lack detailed method descriptions and fail to release related materials (code and datasets).  For example, He & Bao, 2020 labelled a private dataset and trained a supervised model, but they did not public their dataset and did not describe the network structure. These papers lack key information and related materials, which prevents other researchers from re-implementing and following their work.
3. Instead, we will public our circuit simulator, detailed source code and the full dataset, if accepted. For the EDA design community, our algorithm and source code can serve as a strong and fair baseline. For the AI community, our work provides a new evaluation benchmark for the combinatorial optimization problem, and our method also provides some insights on solving other combinatorial optimization problems.

Currently, such work on the EDA-AI boundary is targeted exclusively for AI conferences. This exclusion limits progress for the EDA design. The comments above suggest that our work will help remedy this.

To address the reviewers' questions, we have updated a new version of our paper. We corrected some writing typos and added the wallclock time of different algorithms in the experiment. We'll really appreciate it if the reviewers could improve their scores and accept our new version.

---

### Author Response · Authors · 2020-11-23
**Overall Response to Reviewers' New Comments (new version updated: add experiments on the sequential A* algorithm with more runs)**

We thank reviewers for their new comments.

To address reviewers' questions, we increase the number of runs of the sequential A* baseline, which has competitive running time to our proposed algorithm. Results show that our method still outperforms these baselines.

We tried but were not able to implement a two-stage baseline in the rebuttal period, but will gladly include it by the camera-ready. Even given that, we also point out:
1. The two-stage method is just a routing framework, which divides a complex routing task into two simpler routing stages. Any one-stage routing method, such as the sequential A* or our Ranking Cost algorithm, can be merged into a two-stage routing framework. Moreover, the sequential A*  algorithm is the most commonly used algorithm in current EDA tools for circuit routing. Therefore, we ( and also recent learning-based papers) mainly compare algorithms with the sequential A* algorithm. We think the sequential A*  algorithm can serve as a fair baseline in the circuit routing task.
2. **To best of our knowledge, we are the first learning-based work with state-of-the-art results in this field, which can public source codes**. For the EDA design community, our algorithm and source code can serve as a strong and fair baseline. For the AI community, our work provides a new evaluation benchmark for the combinatorial optimization problem.
3. Our paper proposes a novel method to solve the circuit routing problem (or the vertex-disjoint path problem), which takes advantages of both the search-based algorithm and the learning-based algorithm. And our work also raises some insights on solving other combinatorial optimization problems (e.g., we can also learn a variable order and values using the same formulation in RC for the Max-SAT problem). We hope reviewers can evaluate our work more based on our algorithm.

We'll appreciate it if the reviewers could improve their scores and accept our new version.

---

### Author Response · Authors · 2020-11-24
**[some thoughts about the running time and time complexity]**

Since reviewers pay more attention to the time complexity of our RC algorithm and the sequential A* algorithm, we also point out:

1. **The sequential A* algorithm (the A*) suffers from non-optimal solution problem as stated in Section 2.2.2**. Even given the full sampling, the A* will fail in some cases. For example, in our experiments for maps with the size of 16x16 (4 pairs), there are only 4!=24 orders, and we sampled all the possible orders for the A* since our maximum sampling number for is 200. But the final connected rate of the A* is 0.96. However, our RC algorithm can reach 100% connected rate. **It indicates that our method can achieve the performance which the fully sampled** A* **will never achieve**.
2. A learning-based method needs to train parameters which could take many training steps, but it could be reduced via better optimizers. **Designing a faster optimizer is not the main contribution of our paper. The key contribution of our paper is the Ranking Cost framework, which innovatively combines search-based methods and learning-based methods to form an efficient and trainable router under a proper parameterization**. And any future optimizer can be applied to our framework. However, the number of running steps of the A* is fixed and can not be reduced. We hope reviewers can evaluate our work more based on the algorithm.

At present, the trained cost maps can not generalize to unseen tasks, which means that our RC framework needs to train from scratch for each map. But this could be fixed by designing a better model. For example, we could carefully design a deep neural network, which takes maps as input and outputs cost maps and orders. In this way, the learned knowledge will be stored in the deep model, and it can speed up the training time when encountering new maps. And this will remain as our future work.

---

### Decision · Program_Chairs · 2021-01-07
**Final Decision**

**Decision:**

Reject

**Comment:**

The paper studies an interesting problem motivated by VLSI design. The reviewers agree that there are interesting aspects of the RC algorithm. Nevertheless, the paper could be improved by a clearer characterization/apples-to-apples comparison to baselines, particularly regarding computation cost, use of parallelism, as well as a more thorough contrast to state of the art in general. Given the contribution is experimental, and this is a well studied problem, it is important to establish whether the solution is indeed best-in-class; cost due to training should be taken into account, and minimized to the extend possible. Going beyond the baselines considered here, as well as reviewing possible theoretical connections to other problems and guarantees, would also strengthen the paper.